# Arbuscular Mycorrhizal Fungi Selectively Promoted the Growth of Three Ecological Restoration Plants

**DOI:** 10.3390/plants13121678

**Published:** 2024-06-18

**Authors:** Hengkang Xu, Yuchuan Shi, Chao Chen, Zhuo Pang, Guofang Zhang, Weiwei Zhang, Haiming Kan

**Affiliations:** 1Institute of Grassland, Flowers and Ecology, Beijing Academy of Agriculture and Forestry Sciences (BAAFS), No. 9 Shuguang Garden Middle Road, Haidian District, Beijing 100097, China; xuhengk@163.com (H.X.); xiaoyingwulu@163.com (C.C.); pangzh-84@163.com (Z.P.); zhanggf0727@126.com (G.Z.); zhangwei492@163.com (W.Z.); 2College of Grassland Science and Technology, China Agricultural University, Beijing 100107, China; yuchuan@cau.edu.cn

**Keywords:** AMF, ecological restoration, fungi–plant mutualism, nutrient acquisition, plant productivity

## Abstract

Arbuscular mycorrhizal inoculation can promote plant growth, but specific research on the difference in the symbiosis effect of arbuscular mycorrhizal fungi and plant combination is not yet in-depth. Therefore, this study selected *Medicago sativa* L., *Bromus inermis Leyss*, and *Festuca arundinacea* Schreb., which were commonly used for restoring degraded land in China to inoculate with three AMF separately, to explore the effects of different AMF inoculation on the growth performance and nutrient absorption of different plants and to provide a scientific basis for the research and development of the combination of mycorrhiza and plants. We set up four treatments with inoculation *Entrophospora etunicata* (EE), *Funneliformis mosseae* (FM), *Rhizophagus intraradices* (RI), and non-inoculation. The main research findings are as follows: the three AMF formed a good symbiotic relationship with the three grassland plants, with RI and FM having more significant inoculation effects on plant height, biomass, and tiller number. Compared with C, the aboveground biomass of *Medicago sativa* L., *Bromus inermis Leyss*, and *Festuca arundinacea* Schreb. inoculated with AMF increased by 101.30–174.29%, 51.67–74.14%, and 110.67–174.67%. AMF inoculation enhanced the plant uptake of N, P, and K, and plant P and K contents were significantly correlated with plant biomass. PLS-PM analyses of three plants all showed that AMF inoculation increased plant nutrient uptake and then increased aboveground biomass and underground biomass by increasing plant height and root tillering. This study showed that RI was a more suitable AMF for combination with grassland degradation restoration grass species and proposed the potential mechanism of AMF–plant symbiosis to increase yield.

## 1. Introduction

Arbuscular mycorrhizal fungi (AMF) are an endophytic mycorrhizal fungus that is one of the most important members of the soil microbial community and can form symbiotic relationships with over 80% of terrestrial plants [1,2]. The finely branched structures formed within the root cortical cells of host plants by AMF are called arbuscules, which are the site of bidirectional transport [3]. AMF obtain photosynthetic carbohydrate/carbon from the host plant roots to maintain survival, while they provide more nutrients and water in the soil to the host plant through their extended hyphal network [4]. AMF are one of the major microbial groups that are considered as biofertilizers [5]. Numerous studies have found that AMF can effectively improve the physical and chemical properties of soil, promote the absorption and utilization of various mineral nutrients and water required for plant growth and development [6,7], and enhance plant tolerance to biotic and abiotic stresses [8]. Research has shown that as AMF further invade the root system, the rhizosphere range expands, and the absorption root lifespan extends, thereby affecting plant growth and development [9]. Therefore, AMF inoculation plays an important role in promoting plant productivity through increasing nutrient uptake.

In recent years, researchers have conducted inoculation experiments on different plants using different AMF to determine their infection status and growth indicators and have found different effects. Inoculation with AMF can increase the yield of major food crops such as corn (*Zea mays* L.), wheat (*Triticum aestivum*), and soybean (*Glycine max* L.), improve crop nutrition, and enhance crop resistance to drought, salt, and alkali stress [10,11]. The co-application of AMF and chemical fertilizers can improve the quality of rice and increase the content of iron (Fe) and Zn in plants [12]. Alotaibi et al. found that inoculation with AMF alleviated the toxicity of acidic soil on plant photosynthesis, and barley inoculated with AMF increased photosynthesis by 100% compared to the treatment without AMF [13]. AMF can absorb nutrients from soil through hyphae, increasing the absorption of Ca, Cu, Zn, Mn, and Fe by plants in acidic leaching soil [14]. However, certain AMF have promoting effects on the formation of symbiotic relationships with certain plants, while others have little effect, indicating that there may be selectivity between AMF and plants. For example, in the inoculation experiment of *Artemisia frigida Willd.* and *Leymus chinensis* (*Trin*.) *Tzvel*., it was found that AMF promoted the establishment of seedlings of the latter, while it inhibited the seedling growth of the former [15]. Therefore, it is very meaningful to select a variety of AMF for inoculation experiments with a variety of plants, because it can screen out good symbiotic combinations. Specifically, we sought to address the following questions: (1). Will the experimental AMF used in this study cause different effects on the experimental plants in this study by being selective to the host? (2). Can an assessment model be calculated so that AMF have a similar promotion mechanism for biomass and nutrient uptake for all three test grasses?

Different AMF species have different symbiotic relationships with plants, which may be due to different root characteristics or functional groups (legumes and grasses) of different plants. Therefore, in this study, we chose three degraded grassland restoration grasses for experimentation: *Medicago sativa* L. is a perennial herbaceous plant belonging to the *Fabaceae* family, *Trifolieae* tribe, and is one of the most widely grown forage legumes in the world [16]. Due to its high grass yield, strong adaptability, and rich protein and mineral content, it is known as the “king of forage” [17]. Planting alfalfa can modify soil conditions and improve soil nutrient utilization by promoting the proliferation of beneficial microbiota groups and thus repair degraded soil [18]. *Bromus inermis Leyss* is a perennial, C3, cool-season grass species belonging to the *Poaceae* family [19,20]. Due to its high nutritional value, well-developed root system, lush growth, and wide adaptability to climate conditions, it has been widely used as a forage and pasture crop, for erosion control, phytoremediation, the reclamation of open mines, and as a soil stabilizer [21]. *Festuca arundinacea* Schreb. is a major cool-season perennial grass grown globally [22]. It is widely utilized as turf and forage in temperate regions due to its resistance to heat, drought, and wear [23]. We assumed that AMF were selective towards plants and there was the same mechanism applied for *Medicago sativa* L., *Bromus inermis Leyss*, and *Festuca arundinacea* Schreb. infected with AMF. Inoculation with AMF promotes nutrient absorption in plants, affecting plant height and tiller number both above- and underground, thereby promoting biomass. To address these issues, based on the role of AMF in the restoration of degraded ecosystems, this study selected three commonly used grass species for the restoration of degraded land in China to explore the effects of different AMF inoculation on the growth performance and nutrient absorption of the three plants.

## 2. Materials and Methods

### 2.1. Experimental Materials

The experimental plants were three commonly used plants for vegetation restoration in Beijing’s wasteland: *Medicago sativa* L. (*Medicago*), *Festuca arundinacea* Schreb. (*Festuca*), and *Bromus inermis Leyss* (*Bromus*). The three tested strains of arbuscular endophytes were all from the Bank of Glomales in China (BGC) of the Beijing Academy of Agricultural and Forestry Sciences: *Entrophospora etunicata* (EE, BGC NM03F, 1511C0001BGCAM0041), *Rhizophagus intraradices* (RI, BGC BJ09, 1511C0001BGCAM0042), and *Funneliformis mosseae* (FM, BGC BJ04A, 1511C0001BGCAM0065). The inocula used in the experiment were the above-mentioned strains obtained by the potted inoculation propagation of *Sorghum bicolor × sudanense* as the host. The specific operation was to inoculate the AMF in the prepared culture medium and then sow the plant seeds after germination. When the grass grew to the harvest stage, the aboveground part was removed, and the remaining root and matrix parts were air-dried, crushed, and mixed to obtain the expanded and propagated cluster mycorrhizal fungal agent. The inoculant was mixed filtrates containing fungal spores, mycelium, and other reproductive bodies. The in situ soil of degraded wasteland was used as the culture soil for the experiment, taken from the Yanqing Ecological Station of the Institute of Grassland, Flowers and Ecology, Beijing Academy of Agriculture and Forestry Sciences (BAAFS). The in situ soil was sieved through a 2 mm soil sieve and then subjected to 30 kGy irradiation sterilization.

### 2.2. Experimental Design

The experiment was conducted in a greenhouse of BAAFS. The temperature was 24 ± 5 °C, the humidity was 60~70%, and the lighting time was 13–15 h. *Medicago*, *Festuca*, and *Bromus* were inoculated with GE, GM, and GI, respectively, along with a non-inoculation treatment (C), for a total of 12 treatments, with 4 replicates in each treatment. Add 1kg of sterilized soil and 100 g of bacterial filtrate to the potted plants inoculated with AMF. For C treatment, add an equal amount of sterilized in situ soil, sterilized inoculum, and bacterial filtrate (50 mL), to maintain microbial community consistency.

Uniform sized seeds (n = 3) of the experimental species were sterilized using a 10% H_2_O_2_ solution (SCRC, Shanghai, China). Then, rinse them several times with sterile water, evenly spread them in a Petri dish containing moist filter paper, and place them in an incubator for germination. Select four plant seedlings with similar growth, and transplant them into pots, then cover them with 1 cm soil, and spray water. After being completely randomly arranged and cultured in a greenhouse for 2 weeks, 2 plants were retained. After AMF formed stable fungal connections in the soil, plants were regularly watered and supplemented with Hoagland’s nutrient solution during planting to ensure normal growth and development. After 90 days of cultivation, harvest the plants.

### 2.3. Experimental Method

Measurement of plant growth indicators. Before harvest, measure the height of the plant and the number of tillers. After harvest, rinse the plants with distilled water, dry them at 105 °C for 0.5 h and 80 °C for 48 h, and measure the aboveground and underground biomass, respectively. AMF mycorrhizal infection. Select three plant roots randomly. Wash the roots multiple times with sterile water to remove surface mud and dirt, and cut them into 1.0–2.0 cm root segments. The mycorrhizal fungal infection rate was measured according to the method of Phillips and Hapman [24]. After the soil sample was air-dried, the AMF spores were separated using the method of Koske et al. [25]. The spores were counted under a stereomicroscope, and the spore density was calculated as the spore content per gram of air-dried soil. Measurement of mycelial density using filtration method [26]. Place the filter membrane on a glass slide, add 3 drops of 0.05% Trypan blue staining solution, air-dry, and observe under a microscope.

Determination of plant nutritional indicators. The total nitrogen (TN) was measured with a C/N analyzer (Rapid CS Cube, Elementar, Langenselbold, Germany). The total phosphorus (TP) was measured by the sodium hydroxide melting–molybdenum barium colorimetric method [27]. The total (TK) was measured by ammonium acetate extraction–atomic absorption spectrophotometry [28].

### 2.4. Data Analysis

Experiment data were analyzed using a one-way analysis of variance (ANOVA) by Duncan’s multiple range test and SPSS 20.0 software for processing; the significance level was *p* < 0.05. Before analysis, the Shapiro–Wilk test was used to evaluate the normality of the data. Levene’s test was used for the analysis of variance to determine the homogeneity of variance. The relationship between nutrient content (N, P, and K) and its stoichiometric ratio and biomass was analyzed by linear regression. Partial least squares path modeling (PLS-PM) was performed using the packages of “plspm” in R (https://www.r-project.org/, Version v.4.1.2) to statistically explore how mechanistic pathways in plants inoculated with AMF affect plant aboveground and underground biomass. All the figures were created in R 4.1.2.

## 3. Results

### 3.1. AMF Mycorrhizal Infection

The infection effects of RI and FM on the three plants were all significantly better than those of EE (*p* < 0.05). The infection rate was RI > FM > EE (Figure 1a, *p* < 0.05). The spore density of FM and RI showed no significant difference (Figure 1b, *p* > 0.05) and were significantly higher than that of EE, except for *Medicago*. The hyphal density of the three plants was significantly higher in RI and FM than in EE (Figure 1c, *p* < 0.05).

### 3.2. Plant Growth

There was a significant difference between all treatments inoculated with AMF and C in aboveground biomass, underground biomass, the number of tillers, and height (Figure 2, *p* < 0.05), except for Festuca inoculated by EE (belowground biomass) which was non-significantly different from the control (*p* > 0.05). Among the indicators of *Medicago* and *Festuca*, EE treatment was significantly lower than RI and FM treatments (*p* < 0.05), but there was no significant difference in the indicators after *Bromus* inoculation with the three AMF (*p* > 0.05).

### 3.3. N, P, and K Absorption and Stoichiometry

There was no significant change in the shoot N content of *Bromus* and *Festuca* for four treatments (Figure 3a, *p* > 0.05); The *Medicago* shoot N content was as follows: EE > FM > EE > C. For the shoot P and K of the three plants, there was a significant increase after inoculation with AMF compared with C. Among them, RI was the most significant for raising the shoot P and K of the three plants, except EE was more significantly effective for the shoot K of *Festuca* (Figure 3b,c, *p* < 0.05).

The root N content of *Bromus* after inoculation with AMF was RI > FM > C > EE (Figure 4a); for *Festuca* and *Medicago*, EE and FM treatments were significantly higher than C treatment (*p* < 0.05). The content of root K, FM, and EE treatments was significantly higher than that of C (Figure 4b, *p* < 0.05). The root K content of AMF treatments was significantly higher than C, and RI was the most significant (Figure 4c, *p* < 0.05).

### 3.4. Relationship between Nutrient Uptake and Plant Growth

The results showed (Figure 5 and Figure 6) that the biomass of three plants was significantly positively correlated with the absorption of P and K. Correlation analysis showed that the aboveground biomass of *Bromus* was significantly correlated with shoot P (Figure 5b, R^2^ = 0.21, *p* < 0.05), while *Festuca* (R^2^ = 0.50, *p* < 0.01) and *Medicago* (R^2^ = 0.48, *p* < 0.01) were extremely significantly correlated with shoot P. In addition, there was a significant correlation between aboveground biomass and shoot K for *Bromus* (Figure 5c, R^2^ = 0.35, *p* < 0.01), *Festuca* (R^2^ = 0.27, *p* < 0.05), and *Medicago* (R^2^ = 0.36, *p* < 0.01).

The underground biomass is significantly correlated with the root P and K of *Medicago* (Figure 6b,c, R^2^ = 0.31, *p* < 0.05) and extremely significantly correlated with the root P and K of *Bromus* (P: R^2^ = 0.44, *p* < 0.01; K: R^2^ = 0.44, *p* < 0.01) and root P of *Festuca* (R^2^ = 0.54, *p* < 0.01).

To determine how plants nutrient uptake affects plant growth characteristics, we performed the PLS-PM analysis (Figure 7). Our results indicated that the nutrient absorption of the shoot and root is the primary driver of plant biomass, with regulation by changing plant height and tiller numbers. After the symbiosis between plants and AMF, the nutrient absorption ability of roots was enhanced, which promoted the tillering of plants and improved the underground biomass. Nutrients are transported to the aboveground through the underground roots, increasing plant height and consequently aboveground biomass. Similar mechanisms were observed in different plants.

## 4. Discussion

### 4.1. Different AMF Have Different Symbiotic Relationships with Different Plants

In this experiment, inoculation with EE, FM, and RI can effectively colonize the three grassland plants and establish a good symbiotic relationship (Figure 1). From the root infection rates of the three grassland plants inoculated with AMF, it can be inferred that RI and FM have a better affinity for the three plants than the EE treatment and can better play a dominant symbiotic role in plant growth. The infection rate of the same or different AMF on different plant roots varies, which may be due to differences in the structural characteristics of different AMF and differences in the selectivity of AMF towards symbiotic plants [29]. Plant species can construct niches for soil organisms and thereby alter the composition of soil biota so that individual plant species influence the community structure of AM fungi in their rhizosphere such that different AM fungi show preference for different plant hosts [30,31]. In addition, Ibijbijen et al. also validated this conclusion, who indicated that in the application of AMF microbial agents, it is necessary to pay attention to screening advantageous AMF for specific host plants, in order to play the maximum role in the plant growth process [32].

### 4.2. Effect of AMF Inoculation on N, P, and K Uptake in Plants

N, P, and K are essential nutrients for plant growth and development, which have regulating effects on plant growth and are also important factors limiting plant growth [33,34]. The three AMF promoted the absorption of N, P, and K of the three plants and increased the contents of N, P, and K in plants, among which the promoting effects on P and K were more significant (Figure 3 and Figure 4). One of the reasons why AMF can promote the absorption of nutrients is that AMF can expand the absorption area of plant roots. Ames et al. confirmed by using the ^15^N-labeled method that AMF mycelium could absorb NH_4_^+^ in soil several centimeters outside the roots and transport it to plant roots [35]. About one-third of the N of plant root proteins is supplied by AMF which have a strong absorption and transport capacity [36]. Another reason is that AMF improve the availability of soil N, P, and K [37]. Nitrogen ions extracted from the soil by the AMF mycelium are converted into arginine and transported in this form through the mycelium to the host root [38]. And it can absorb N from complex organic N and accelerate its degradation process [39,40]. AMF stimulate the root system to secrete organic acids to dissociate insoluble phosphate, thus promoting the plant’s absorption of P [41]. Liu et al. identified a mycorrhizal specific K transporter SlHAK10 (Solanum lycopersicum High-activity K Transporter10) in the study of tomato (*Solanum lycopersicum*) [42], and SlHAK10 overexpression can promote K^+^ absorption in plants.

Plant N/P can be used as an indicator to judge the nutrient supply status of soil to plant growth and can represent the restriction effect of N or P on plant growth [43]. Braakhekke and Hooftman pointed out in an experimental study on grassland that when N/P < 14, plants are limited by N; when N/P > 16, plants are limited by P, and the ratio between the two is limited by both N and P [44]. The results of this experiment showed that the N/P ratio of three grassland plants in un-inoculated treatment was lower than 14, indicating that vegetation growth in degraded land was more inclined to be restricted by N in the seedling stage. We found that inoculation with EE and FM significantly increased root N content in three plants (Figure 4a), and the root N/P of *Medicago sativa* L. was significantly increased in particular (Figure 4d), which indicates that AMF break the N limit of degraded soil. We also found that the N content of *Medicago sativa* L. inoculated with AMF was higher than that of other two gramineous plants (Figure 3a and Figure 4a). This is due to the fact that rhizobium symbiosis can improve the nitrogen fixation ability of legumes compared with other gramineous plants [45]. AMF and rhizobia form an AMF–rhizobia–plant symbiosis system with legumes, which increases the nodulation number of rhizobia and provides sufficient N sources for the growth of AMF mycelia and improves the infection rate.

### 4.3. Effects of AMF Inoculation on Plant Aboveground and Underground Biomass

Biomass is one of the important indicators to measure the growth status of plants, and it is the embodiment of material accumulation in the process of plant growth and metabolism [46]. In this study, the plant biomass, height, and numbers of tillers of the three plants inoculated with AMF were all improved to varying degrees (Figure 2), which was similar to the results of other scholars [47,48], indicating that AMF inoculation had a significant promoting effect on the growth of *Medicago*, *Bromus*, and *Festuca*. Liu et al. also found that licorice (*Glycyrrhiza uralensis Fisch*) inoculated with AMF significantly increased plant height, fresh weight, dry weight, root length, and lateral root number [49], which can also prove that AMF inoculation has a promoting effect on plant biomass. In addition, correlation analysis showed that P and K were positively correlated with the three plants’ biomass, indicating AMF promoted the absorption of P and K in plants, thereby increasing the biomass (Figure 5 and Figure 6). The results of this study confirmed that after AMF formed a symbiotic relationship with host plants, a huge mycelium network was formed around AMF, which expanded the area of water and nutrients absorbed by plant roots in soil, improved the utilization rate of nutrients in soil, and promoted the growth and development of plants [50].

In addition, the RI infection rate of the three plants was the highest, and the plant height, biomass, and tillering number under this treatment were also higher than those under other AMF treatments (Figure 1a and Figure 2), indicating that the infection rate of AMF was significantly positively correlated with the growth promotion effect of the plants. However, in this experiment, it was found that after *Bromus* was inoculated with EE, the infection rate was significantly lower than that under FM treatment, but the number of tillers and aboveground biomass were higher than that under FM treatment (Figure 1a and Figure 2a,c), which was similar to the results of Enkhtuya et al. [51]. The promoting effect of inoculation with AMF with a low infection rate was better than that with a high infection rate, which indicates that the infection rate is not the only index to measure the effect of AMF infection on the growth of host plants. It was also found that there were significant differences in plant biomass, height, and tiller number among the three grassland plants inoculated with different AMF (Figure 2, *p* < 0.05). On the one hand, the possible reasons for this result are that AMF have different structural characteristics, so they have different absorption capacities of nutrients and water in soil. On the other hand, AMF have certain selectivity to host plants, and the same AMF has different growth-promoting effects on different host plants. In conclusion, the three AMF promoted the absorption of N, P, and K in *Medicago*, *Bromus*, and *Festuca*. However, the growth promotion effect of AMF was mainly determined by the change in plant growth and nutrient content. The promotion effect of dominant strains on plant growth was significantly higher than that of common strains [52].

## 5. Conclusions

In the present study, we explored the effects of AMF inoculation on the growth and nutrient uptake of three common grassland plants in degraded land in China. AMF had selectivity towards plants, and different AMF had different infection rates towards different plants. AMF had a similar promotion mechanism for the three plants: AMF inoculation increased the nutrient uptake of plants and then increased aboveground biomass and underground biomass by increasing plant height and root tillering, respectively. This study proposed the potential mechanism of AMF and plant symbiosis to increase yield and provided theoretical support for the further development and utilization of dominant plants in the restoration of degraded land. Future research should pay more attention to the optimal combination mode of different plants and different mycorrhizal fungi so as to provide a more optimal inoculation scheme for plant and mycorrhizal symbiosis.

## Figures and Tables

**Figure 1 plants-13-01678-f001:**
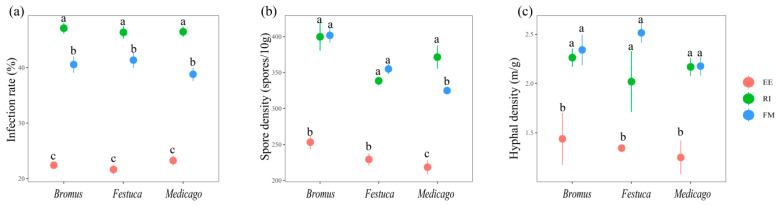
Mycorrhizal infection rate (**a**), spore density (**b**), and hyphal density (**c**) of three grassland plants. *Bromus*: *Bromus inermis Leyss*; *Festuca*: *Festuca arundinacea* Schreb.; *Medicago*: *Medicago sativa* L. Different lowercase letters above columns represent significant differences among these treatments according to Duncan tests.

**Figure 2 plants-13-01678-f002:**
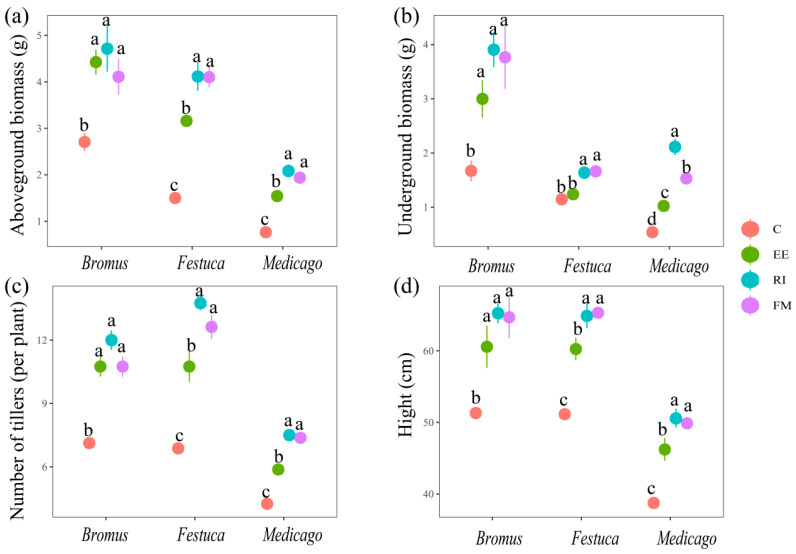
Effects of AMF on aboveground biomass (**a**), underground biomass (**b**), numbers of tillers, (**c**) and plant height (**d**) of three grassland plants. *Bromus*: *Bromus inermis Leyss*; *Festuca*: *Festuca arundinacea* Schreb.; *Medicago*: *Medicago sativa* L. Different lowercase letters above columns represent significant differences among these treatments according to Duncan tests.

**Figure 3 plants-13-01678-f003:**
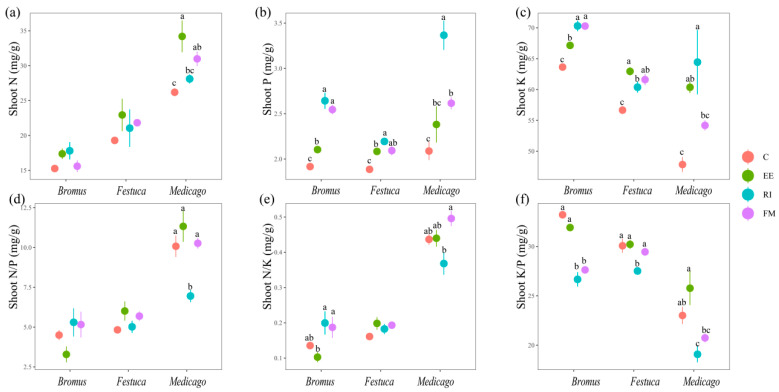
(**a**–**f**) Effects of AMF on total N, P, K, N/P, N/K, and K/P of shoot of three grassland plants. *Bromus*: *Bromus inermis Leyss*; *Festuca*: *Festuca arundinacea* Schreb.; *Medicago*: *Medicago sativa* L. Different lowercase letters above columns represent significant differences among these treatments according to Duncan tests.

**Figure 4 plants-13-01678-f004:**
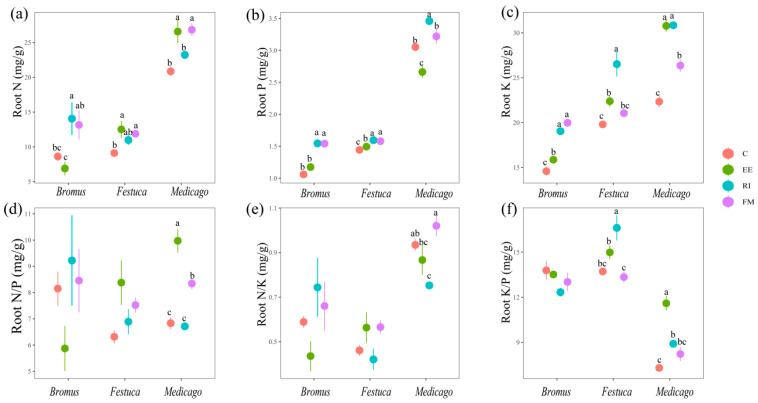
(**a**–**f**) Effects of AMF on total N, P, K, N/P, N/K, and K/P of root of three grassland plants. *Bromus*: *Bromus inermis Leyss*; *Festuca*: *Festuca arundinacea* Schreb.; *Medicago*: *Medicago sativa* L. Different lowercase letters above columns represent significant differences among these treatments according to Duncan tests.

**Figure 5 plants-13-01678-f005:**
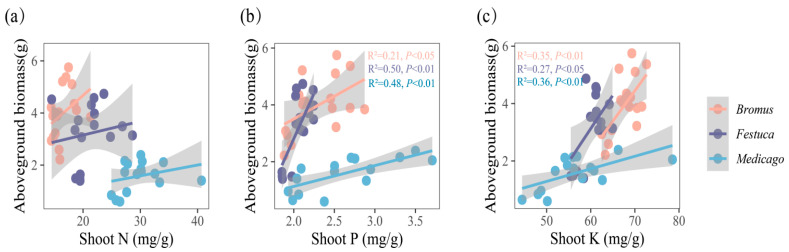
(**a**–**c**) Relationships between aboveground biomass and shoot nutrient uptake (N, P, K, N/P, K/P, N/P) for three plants. *Bromus*: *Bromus inermis Leyss*; *Festuca*: *Festuca arundinacea* Schreb.; *Medicago*: *Medicago sativa* L. R^2^ represents the judgment coefficient, which is used to measure the explanatory power of the regression equation for y. Statistically significant correlations are indicated with *p* < 0.05, *p* < 0.01, and *p* < 0.001.

**Figure 6 plants-13-01678-f006:**
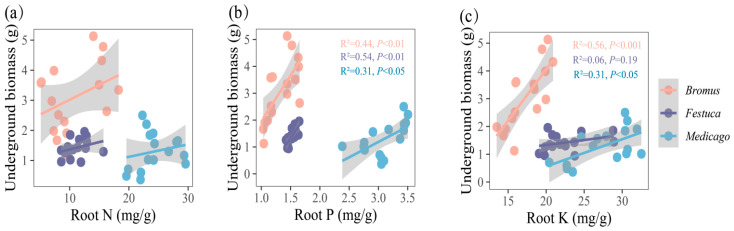
(**a**–**c**) Relationships between underground biomass and root nutrient uptake (N, P, K, N/P, K/P, N/P) for three plants. *Bromus*: *Bromus inermis Leyss*; *Festuca*: *Festuca arundinacea* Schreb.; *Medicago*: *Medicago sativa* L. R^2^ represents the judgment coefficient, which is used to measure the explanatory power of the regression equation for y. Statistically significant correlations are indicated with *p* < 0.05, *p* < 0.01, and *p* < 0.001.

**Figure 7 plants-13-01678-f007:**
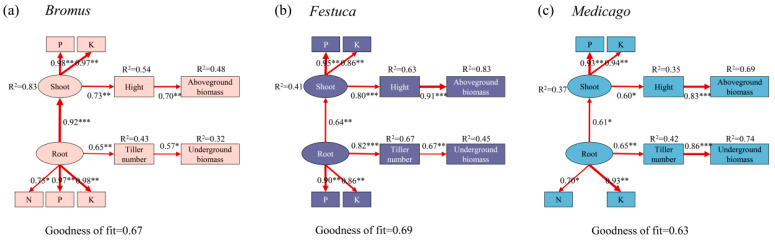
Partial least squares path models (PLS-PM) reveal the direct and indirect effects of plant nutrient uptake on plant growth characteristics. *Bromus*: *Bromus inermis Leyss*; *Festuca*: *Festuca arundinacea* Schreb.; *Medicago*: *Medicago sativa* L. Models are evaluated using goodness of fit (GOF). The red arrows represent significant path relationships, and the width of arrows is proportional to the standardized path coefficient. Values beside the variables represent the coefficients of determination. Asterisks indicate the statistical significance (* *p* < 0.05, ** *p* < 0.01, *** *p* < 0.001).

## Data Availability

Data will be made available on request. The data are not publicly available due to privacy.

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
