# Peer review of "Arbuscular Mycorrhizal Fungi Selectively Promoted the Growth of Three Ecological Restoration Plants"

_plants, 2024, doi:10.3390/plants13121678_

Round 1
Reviewer 1 Report
Comments and Suggestions for Authors
The choice of topic of the manuscript has both scientific topicality and the possibility of economic utilization. The manuscript is easy to read, and the editing is good with the proportional division of the chapters. The objectives of the research are clearly presented in the introduction chapter. The design of the experiments and the methods meet the scientific expectations of our time. The figures used to explain the results is basically beautiful and transparent, but it is burdened with some shortcomings. The availability and comprehensibility of the data could be made more complete with a supplement file. The discussion was born out of substantial and thorough work. The literature is rich and cites many recent papers.
Suggestions for improvements and changes:
The current scientific names of the three AMFs used/included in this research (Index Fungorum - Search Page):
Glomus etunicatum: Entrophospora etunicata (W.N. Becker & Gerd.) BÅ‚aszk., Niezgoda, B.T. Goto & Magurno
Glomus mosseae: Funneliformis mosseae (T.H. Nicolson & Gerd.) C. Walker & A. Schüßler
Glomus intraradices: Rhizophagus intraradices (N.C. Schenck & G.S. Sm.) C. Walker & A. Schüßler
Ad p. 2.: Medicago sativa is not a grass species ("..promotion mechanism in the three test grass species?" ; "three different types of degraded grassland restoration grass..." M
ad 3.3 "There was no significant change in the shoot N content..." Since the biomass of the mycorrhized plants is greater than that of the control, it would be worthwhile to evaluate the N, P and K contents for whole plants as well (at least in text) due to the dilution effect. It is conceivable that in this case also significant differences would arise in the case of N (see Fig. 3/a).
Ad Fig. 5. & Fig. 6. Many figures lack R2 values and statistical evaluation.
ad 4.2 Correct: Braakhekke and Hooftman
ad 4.2 The last sentence of the chapter was left without a reference.
ad 4.3 Correct: …Enkhtuya et al.[51].
ad 5. Instead of "AMF had the same promotion mechanism", it is suggested: "...similar promotion...)" ?
Unfortunately, relatively large number of formal errors can be found:
- Despite the provisions of the Botanical Nomenklatura, the writing of genus names and species names was often left without emphasis,
- Despite the provisions of the Botanical Nomenklatura, author names were often written in italics,
- ad 2.1 "in situ" / foreign language excerpt, writing in italics is recommended,
- Missing spaces (p.10.),
- ad References /19.: correctly: Northern
Comments on the Quality of English Language
The MS's use of English corresponds to what is expected of academic theses, however, according to the reviewer, the text can still be improved in the case of some sentences.
Author Response
Comments 1: The current scientific names of the three AMFs used/included in this research (Index Fungorum - Search Page). |
Response 1: Thank you for pointing this out. We agree with this comment. Therefore, we have modified the Latin scientific names in the abstract and the second part, and the abbreviations did not be modified.
|
Comments 2: Ad p. 2.: Medicago sativa is not a grass species ("..promotion mechanism in the three test grass species?" ; "three different types of degraded grassland restoration grass..." M |
Response 2: Agree. We have correspondingly deleted "type" and only emphasized that they are three experimental grass.
Comments 3: ad 3.3 "There was no significant change in the shoot N content..." Since the biomass of the mycorrhized plants is greater than that of the control, it would be worthwhile to evaluate the N, P and K contents for whole plants as well (at least in text) due to the dilution effect. It is conceivable that in this case also significant differences would arise in the case of N (see Fig. 3/a). Response 3: Thank you for your prompt. Our goal is to focus on the nutrient concentration differences between the aboveground and underground parts after discovering differences in biomass.
Comments 4: Ad Fig. 5. & Fig. 6. Many figures lack R2 values and statistical evaluation. Response 4: Thank you for your reminder. The absence of R2 values and statistical evaluation in the graph indicates that the differences are not significant. Markers only appear when there is a significant difference or correlation between the two.
Comments 5: ad 4.2 Correct: Braakhekke and Hooftman Response 5: Thank you for your correction. We have corrected and highlighted the text in yellow.
Comments 6: ad 4.2 The last sentence of the chapter was left without a reference. Response 6: Thank you very much for your correction. We have added it to the 52nd reference.
Comments 7: ad 4.3 Correct: …Enkhtuya et al.[51]. Response 7: Thank you for your correction. We have corrected and highlighted the text in yellow.
Comments 8: ad 5. Instead of "AMF had the same promotion mechanism", it is suggested: "...similar promotion...)" ? Response 8: Thank you for your reminder. We have changed "same" to "similar".
Comments 9: Despite the provisions of the Botanical Nomenklatura, the writing of genus names and species names was often left without emphasis, Response 9: Agree. We have corrected all parts of the text that involve Latin scientific names, species names, and genus names to italics and highlighted them in yellow.
Comments 10: Despite the provisions of the Botanical Nomenklatura, author names were often written in italics Response 10: Thank you for pointing this out. According to the Binary nomenclature, the name and abbreviation of the namer should be in standard character, not italics.
Comments 11: ad 2.1 "in situ" / foreign language excerpt, writing in italics is recommended Response 11: Thank you for your correction. We have corrected the text to italics and highlighted it in yellow.
Comments 12: Missing spaces (p.10.) Response 12: Thank you for your correction. We have corrected and highlighted the text in yellow.
Comments 13: ad References /19.: correctly: Northern Response 13: Thank you for your correction. We have corrected and highlighted the text in yellow. |
4. Response to Comments on the Quality of English Language |
Response: Thank you for your valuable and thoughtful comments. We have carefully checked and improved theEnglish writing in the revised manuscript. |

Reviewer 2 Report
Comments and Suggestions for Authors
Please see attached

Author Response
Comments 1: Line: crops such as corn, wheat (Triticum aestivum), and soybean (Glycine max L.),. Comment: provide scientific name for corn in brackets as per wheat and soybean. Response 1: Thank you for your suggestion. We have added the Latin scientific name of maize (Zea mays L.) in brackets and highlighted it in yellow in line 51.
Comments 2: Where et al is provided for reference, this should be italicized and should ne in the format “et al.,”. Verify and amend throughout the manuscript Response 2: We agree with this comment. We have Verified and amended throughout the manuscript.
Comments 3: Research questions should be better shaped. The first is a double question. Restructure. The second is a bit cumbersome and should be improved. Perhaps, remove questions and provide clear aims? Response 3: Agree. We have made modifications and integrated scientific questions. “1. Will the experimental AMFs used in this study cause different effects on the experi-mental plants in this study by being selective to the host? 2. Can an assessment model be calculated so that AMF has the similar promotion mechanism about biomass and nutrient uptake for all three test grasses?”
Comments 4: All numbers and units, included % and centigrade, must be separated by a space as in accordance to the internation system of measurement (SI) Response 4: Thank you for your reminder. We have reviewed and revised the entire manuscript and highlighted it in yellow.
Comments 5: All Latin and Latin-derived words shall be italicized (in vivo, in vitro, etc., and so forth). Response 5: Thank you for your reminder. We have reviewed and revised the entire manuscript and highlighted it in yellow.
Comments 6: The sentence structure throughout the manuscript needs to be improved for clarity purposes. In some case appear a literary translation from the native language. Response 6: We apologize for the poor language of our manuscript. We have now worked on both language and readability and have also involved English speakers for language corrections. We really hope that the flow and language level have been substantially improved.
Comments 7: The three tested strains were all from the Bank. Should be changed to: The three tested strains of arbuscular endophytes were all from the Bank… Response 7: We appreciate it very much for this good suggestion, and we have done it according to your ideas in line 101.
Comments 8: The sentence; “The inocula used in the experiment were the abovementioned strains obtained by potted propagation of the Sorghum bicolor × sudanense as the host” is unclear and should be improved. Response 8: Agree. We have revised and specified the original sentence to: “The inocula used in the experiment were the abovementioned strains obtained by potted inoculation propagation of the Sorghum bicolor × sudanense as the host. The specific operation was to inoculate the AMFs in the prepared culture medium, and then sow the plant seeds after germination. When the grass grew to the harvest stage, the above-ground part was removed, and the remaining root and matrix parts were air-dried, crushed and mixed to obtain the expanded and propagated cluster mycor-rhizal fungal agent.”in line 104-110.
Comments 9: The sentence:” the inoculants were bacterial filtrates mixed with fungal spores, mycelium, and other reproductive bodies” states bacterial filtrates as the species tested are fungi, where are the bacteria coming from? Naturally occurring species? Is so, a few lines are needed to clarify this. Response 9: Sorry, this is our expression error. The inoculum used in this experiment is a mixed substrate containing fungal spores, hyphae, and other reproductive bodies. We have made modifications in line 111.
Comments 10: The sentence: “The plant culture soil was the in-situ soil of degraded wasteland, taken from” should be changed to: In-situ soil of degraded wasteland was used as the culture soil for the experiment. Response 10: Thank you for your modification. We have adopted it and highlighted it in yellow in the text in line 112.
Comments 11: The sente: The three plants in the experiment concluded four treatments with four replications respectively, has a few issues, what are the 3 plants? Are the authors saying the 3 species? Rather than concluded, included is the word. Response 11: Thank you for your suggestion. We have changed the statement to: “Medicago, Festuca, and Bromus were inoculated with GE, GM, and GI respectively, along with a non-inoculation treatment (CK), for a total of 12 treatments, with 4 replicates in each treatmentand” in line 118-121.
Comments 12: Control should be indicated with the letter C as opposed to CK. Please ensure to review and amend the manuscript and all figures (……) Response 12: Thank you for your comment. We have revised and reviewed the entire article with the letter C as opposed to CK.
Comments 13: The sentence “Use a 10% H2O2 (SCRC, Shanghai, China) solution to sterilize the surface of three plants seeds with uniform size and full particles” needs better structure. For example: Uniform sized seeds (n=3) of the experimental species were sterilized using a 10% H2O2 solution. Response 13: Thank you for your modification. We have adopted it and highlighted it in yellow in the text in line 125-126.
Comments 14: “Hoagland's nutrient solution during planting to ensure normal growth and development. After 90 days of cultivation, harvest the plants.” It is unclear and, arises a key concern on the experimental design, the use of Hoagland’ solution. If the purpose of this study was to evaluate AME, Hoagland nutrients would partially invalidate the impact of AME. The authors will need to be careful in making a strong argument on the use of this nutrient supplementation. Response 14: We will be happy to deeply discuss the addition of Hoagland's nutrient solution, based on helpful comments from the reviewers. Hoagland's nutrient solution is a plant nutrient solution, and the concentration we used is not high, only for the normal growth of plants during the seedling stage. Moreover, nutrient solution was added to all treatments to ensure the singularity of experimental differences. The difference in biomass in the later stage still reflects the effectiveness of AMF inoculation, so this nutrient solution does not affect the accuracy of the results.The study of Hoagland's nutrient solution on barley can prove our claim(Khursheed, M. Q., Salih, Z. R., & Saber, T. Z. Response of Barley (Hordeum vulgare L.) Plants to Foliar Fertilizer with Different Concentrations of Hoagland Solution.)
Comments 15: Figure 1. unreadable. Font is too small, increased this for clarity. Not possible to distinguish the letter codes thus not possible to make sense of the figure. Letter codes at any rate, should be presented in the figure legend. Response 15: We apologize for any inconvenience caused to your reading due to our negligence. We have made corresponding modifications to all the figures in the article regarding this matter.
Comments 16: There were 4 treatments, including the control, why are represented only 3? Response 16: Because the control treatment was not inoculated with arbuscular mycorrhizal fungi, so the infection rate was not calculated and displayed.
Comments 17: Genus names should also be in Italic font. Response 17: We are very sorry, this was our negligence. We have made full revisions and checked.
Comments 18: The Sentence: ?” .” is rather cumbersome and need clarity. except for there was no significant difference between the underground biomass of Festuca inoculated by GE and that of CK (P>0.05) For example: except for Festuca inoculated by GE (belowground biomass) which was non-significantly differ from the control (P>0.05). Response 18: Thank you for your suggestion. We have completed it in line 177-179.
Comments 19: Figure 2. Latin names are to be italicized. Colour coding should be improved as still barely readable. Figure legend is repeating the species. Remove duplications. Response 19: We apologize for any inconvenience caused to your reading due to our negligence. We have made modifications to the repeated legends and overly small letters of all figures in the article.
Comments 20: Figure 3 is once again difficult to examining. All coding and names are required to be increased to be readable. The same goes for figure 4. Response 20: We apologize for any inconvenience caused to your reading due to our negligence. We have made corresponding modifications to all the figures in the article regarding this matter.
Comments 21: Figure 5 and 6. Not entirely sure about the value of those correlational analyses. There is no surprise that shoot biomass is correlated with N/P. Here the authors need a link to both Hoagland supplementation and the AME. Response 21: We want to investigate whether the biomass of plants is related to the increased NPK content of plants inoculated with arbuscular mycorrhizal fungi, so we conducted this correlation analysis. However, due to the unclear correlation between stoichiometry and biomass, we have removed the section on stoichiometry.
Comments 22: Figure 7 is also a questionable figure. No surprise in that and thus it questions what is the value of this analysis. Strongly recommended the removal from the main document and supply as, perhaps, a supplementary-not truly needed though. Response 22: We politely question the suggestion. Figure 5&6&7 were to answer our second scientific question: in addition to some content measurements, we also want to explore the correlation between these factors and calculate a promotion mechanism model applicable to both AMFs and plants in the experiment.
Comments 23: “ In addition, Ibijbijen et al. also validated this conclusion in different bean inoculation experiments, which results indicate that in the application of AMF microbial agents, it is necessary to pay attention to screening advantageous AMF for specific host plants, in order to play the maximum role in plant growth process” it is a rather cumbersome paragraph. Please, edit this carefully to improve readability and clarity Response 23: Thank you for your comment. We have revised and reviewed it in line 266-267.
|
4. Response to Comments on the Quality of English Language |
Response : Thank you for your valuable and thoughtful comments. We have carefully checked and improved theEnglish writing in the revised manuscript. |

Round 2
Reviewer 2 Report
Comments and Suggestions for Authors
The authors improved their manuscript according to the suggestion.
I have no further comments on the manuscript.